# The Impact of Antenatal Balanced Plate Nutrition Education for Pregnant Women on Birth Weight: A Cluster Randomised Controlled Trial in Rural Bangladesh

**DOI:** 10.3390/nu14214687

**Published:** 2022-11-05

**Authors:** Morseda Chowdhury, Camille Raynes-Greenow, Patrick Kelly, Neeloy Ashraful Alam, Kaosar Afsana, Sk Masum Billah, Michael John Dibley

**Affiliations:** 1Health Nutrition and Population Programme, BRAC, BRAC Centre, 75 Mohakhali, Dhaka 1212, Bangladesh; 2Faculty of Medicine and Health, Sydney School of Public Health, The University of Sydney, Camperdown, NSW 2006, Australia; 3James P Grant School of Public Health (JPGSPH), BRAC University, 28 Mohakhali, Bir Uttom A. K. Khandakar Road, Dhaka 1213, Bangladesh; 4Maternal and Child Health Division, International Centre for Diarrhoeal Disease Research, Bangladesh (icddr,b), Dhaka 1212, Bangladesh

**Keywords:** low birth weight, maternal nutrition education, food demonstrations, dietary diversity, cluster RCT, Bangladesh

## Abstract

Low birth weight (LBW) is a global public health problem with the highest prevalence in South Asia. It is strongly associated with maternal undernutrition. In South Asia, intra-household food distribution is inequitable, with lower dietary adequacy in women. Evidence that nutrition education improves diet during pregnancy and reduces LBW is weak. We assessed the impact of nutrition education for pregnant women on birth weight in rural Bangladesh. We conducted a parallel, two-arm, cluster-randomised controlled trial, with 36 clusters allocated equally to intervention (*n* = 445) or standard care (*n* = 448). From their first trimester until delivery, intervention participants received education about eating balanced meals to meet daily dietary requirements with diverse food groups. The primary outcome of mean birth weight was 127.5 g higher in the intervention compared to control women, and the intervention reduced the risk of LBW by 57%. Post hoc analyses showed a significantly higher birth weight and a greater reduction in LBW amongst adolescent mothers. The mean number of food groups consumed was significantly higher in the intervention from the third month of pregnancy than in the control. A community-based balanced plate nutrition education intervention effectively increased mean birth weight and reduced LBW, and improved dietary diversity in rural Bangladeshi women.

## 1. Introduction

Birth weight is a powerful predictor of infant survival [1]. Low birth weight, defined as less than 2500 grams (g), is a composite measure of restricted fetal growth and short duration of gestation, major determinants of infant mortality, childhood growth and development, and adulthood well-being [1,2]. The lower the birth weight, the higher the mortality risk [3,4]. Globally, in 2010, there were an estimated 18 million LBW babies, of whom 59% were term but small for gestational age, and 41% were preterm [5]. An estimated 606,500 neonatal deaths occur annually in low and middle-income countries due to small for gestational age, which accounts for 22% of neonatal deaths [6]. South Asia has the highest prevalence of LBW globally (34%) [6]. Bangladesh is ranked fourth in the global burden of LBW [5].

LBW is strongly associated with maternal undernutrition [7], which induces suboptimal placental growth, influencing nutrient delivery, altering fetal hormones, and contributing to restricted development [8]. A World Health Organization study with 111,000 women reported that mothers in the lowest quartile of pre-pregnancy weight carried an elevated risk of IUGR of 2.55 (95% confidence interval (CI): 2.3, 2.7) and LBW of 2.38 (95% CI: 2.1, 2.5) compared to the upper quartile [9]. In India, the odds ratio for LBW was three times greater in severe energy-deficient low BMI groups than in normal BMI [10]. Consistent with the mechanisms and epidemiological evidence is the high prevalence of LBW in rural Bangladeshi women who have a chronic shortfall of food energy throughout pregnancy [11]. In South Asia, intra-household food distribution is inequitable with lower energy share, animal source food, and dietary adequacy in women, regardless of pregnancy [12].

There is little data on nutrition education’s effects on birth weight in low or middle-income countries, although four meta-analyses have assessed impacts. The first, in 1993, found that nutrition education significantly increased the consumption of protein and vitamins and the birth weight of infants by 300 g. However, the studies lacked statistical power or did not report effect size estimates [13]. The second meta-analysis in 2012 found a 105 g (95% CI: 18, 193 g) increase in mean birth weight with nutrition education and counselling in pregnancy. When stratified by the study site, the effect failed to reach significance in low and middle-income countries (152 g; 95% CI: −81, 384 g) [14]. The third meta-analysis in 2014 examined a variety of dietary interventions in pregnancy, including food supplements, fortified foods, dietary counselling and combinations of these interventions [15]. It included the results of only seven studies from low and middle-income countries with different interventions and outcomes. Furthermore, it examined the effects of the interventions on increasing or decreasing birthweight, as many of the studies were in overweight and obese populations. The fourth, a Cochrane review [16], identified two small trials that reported significantly higher birth weight in the intervention group (mean difference 489.8 g, 95% CI: 427.9, 551.6 g). There are no reports of trials of antenatal nutrition education in rural Bangladesh.

Thus, the evidence for nutrition education to improve diet and maternal and neonatal health indicators, including birth weight, remains weak. Most evidence comes from trials with design or analysis limitations [14] and in high-income countries where counselling is done by professional dietitians or nutritionists [15]. There is a critical knowledge gap about antenatal nutrition education’s efficacy and potential to reduce LBW in South Asia, including Bangladesh. Filling this gap can potentially contribute to the World Health Assembly’s global goal of a 30% reduction in the LBW rate by 2025 [17]. Therefore, we investigated the impact of antenatal nutrition education with demonstrations of a nutritionally energy-protein balanced diet on infants’ birth weight in a low-income population.

## 2. Materials and Methods

### 2.1. Study Design and Participants

We conducted a two-arm, parallel, cluster-randomised controlled trial (cRCT) and have previously described the design [18]. In brief, the study was in the rural district of Sherpur, which has a low human development index, a high percentage in the lowest economic quintile (37.1%) compared to the national average (21.7%) [19] and 48% below the poverty line [20]. The Food Security Atlas of Bangladesh classifies Sherpur as having a high/very high household food insecurity level [21]. In 2016 in Sherpur, the neonatal mortality (41/1000 live births) and under-five mortality (59/1000 live births) were higher than the national average (30 and 47/100.0, respectively) [22].

Sherpur has five sub-districts: Jhenaigati (population 160,452), Nakla (189,685), Nalitabari (251,361), Sherpur Sadar (497,179) and Sreebardi (259,648). Pregnant women, self-reporting to a hospital or formal practitioner (doctor, nurse, or Family Welfare Visitor), receive nutrition education as part of antenatal care (ANC). BRAC (a national NGO; formerly known as Bangladesh Rural Advancement Committee) implements a community-based Maternal, Neonatal, and Child Health (MNCH) program characterised by prospective pregnancy surveillance and basic ANC led by community health workers (CHW) known as *Shasthya Kormi.* One *Shasthya Kormi* serves an area of ~10,000 people, along with ten health volunteers (*Shasthya Shebika*). The *Shasthya Kormi* are village women who have basic 10-years schooling and have received training on health, nutrition, and sanitation issues for six weeks, followed by bi-monthly refreshers.

In our study, a cluster was the catchment area of one *Shasthya Kormi,* which provided logistical convenience and minimised intervention spills into the control group. *Shasthya Kormi,* with at least six months of experience in MNCH, were eligible. Participants were pregnant women 15–49 years, with gestation 12 weeks or less, permanently residing in the area, and not expecting to move from the usual residence until birth. We used the established BRAC pregnancy detection system for monthly menstrual surveillance to identify new pregnancies. Women with delayed menstruation for six weeks or more underwent a urine test. We excluded pregnant women with self-reported chronic disease. All women excluded from the study continued to receive standard care and counselling as in the control clusters. A *Shasthya Kormi* visited each confirmed pregnant woman, screened for eligibility, and obtained verbal informed consent. Recruitment took place from 1 October–31 December 2016. 

### 2.2. Randomisation and Masking

Sherpur district is divided into five sub-districts: Jhenaigati, Nakla, Nalitabari, Sherpur Sadar and Sreebardi, containing 16, 19, 25, 49 and 26 clusters, respectively. We used the stratified proportionate to population size sampling method to obtain 4, 5, 7, 13 and 7 clusters, respectively, from each of the five sub-districts. The clusters were dispersed across the sub-districts and sufficiently separated to minimise the risk of contamination. We selected 36 as the maximum number of *Shasthya Kormi* a single BRAC supervisor could oversee. We randomly allocated the treatments to clusters in a 1:1 ratio by conducting a lottery. We wrote each cluster number onto 36 pieces of paper and put them into five different jars, one for each sub-district, based on the cluster selection made in the first step. The treatment sequence alternated between A (intervention) and B (control), starting with A. We invited a blindfolded volunteer not linked to the study to choose the pieces of paper from each of the jars. We have described this process in more detail in the protocol paper [18]. All eligible pregnant women served by one *Shasthya Kormi* received the intervention or standard care per the random assignment.

We could not mask families or the research team to group allocation; however, we blinded researchers assessing primary and secondary outcomes to the study hypotheses. We also used an objective outcome (birth weight) that is less prone to ascertainment bias.

### 2.3. Procedures

Intervention *Shasthya Kormi* received two days of nutrition training, including a balanced diet and counselling techniques. They also practised community counselling skills under supervision and received one-day training on taking consent and data collection. Enrolled participants were visited monthly to provide ANC that included standard nutrition advice. In intervention clusters, balanced plate intervention replaced the standard nutrition advice using a culturally appropriate diet for Bangladeshi pregnant women [23]. This contained seven-food groups (cereal, lentils, animal protein, vegetables, fruits, milk, and oil), giving a daily 2500 kcal energy and essential micro and macronutrients. The balanced plate was a combination of foods in appropriate portion sizes to meet requirements. Before the trial, we conducted focus groups to identify locally preferred foods and dietary restrictions. We pilot-tested the menu to assess community acceptability and the feasibility of incorporating it into ANC services.

We delivered the intervention individually. Initially, participants received information on a balanced diet and the maternal and fetal health benefits. Then, they learned to prepare a “balanced plate” by demonstrating the right proportions of food groups using the prescribed menu [18]. The community health worker displayed different food combinations on the plate with appropriate alternatives, such as replacing meat (an expensive protein source) with cheaper eggs or farm fish. We emphasised foods commonly missed in the regular diet, such as milk, fruits, and coloured vegetables. Women were encouraged to increase their frequency of food intake to three meals and two snacks per day. We used a 250 mL bowl commonly found in most homes to demonstrate measurements. The community health worker invited family decision-makers to observe the demonstration and assist in making a balanced plate. 

Delivering the intervention took 45 min: 10 for individual counselling, 10 for household counselling, and 25 for demonstration. To help recall messages, we provided a pictorial food chart and a written menu [18]. The community health worker delivered eight nutrition messages during the first visit with a practical demonstration. In subsequent visits, they emphasised messages that were less familiar or difficult to remember. They excluded easily recalled messages in the counselling, such as drinking eight glasses of water or cooking with oil. They also dropped the messages if it was clear the women practised the behaviour. As the number of messages decreased, the community health worker reduced the visit time to 25 min. The mean number of intervention visits was 3.4.

Home visits continued until birth or the end of the pregnancy; ANC from other sources was not discouraged. Husbands were encouraged to reallocate family budgets to purchase nutritious food and mothers-in-law to re-distribute family food to increase their daughter-in-law’s share.

In the control group, pregnant women received the same frequency of home and standard nutrition advice except for the balanced plate demonstrations. Standard advice recommended consuming all food groups and taking iron-folic acid and calcium supplements. More details about the intervention and standard care can be found in the published protocol paper [18].

*Shasthya Kormi* collected demographic, socio-economic, and reproductive data at enrolment plus birth weight measurements within 72 h of birth. Families immediately notified births or pregnancy losses (miscarriage, abortion, and stillbirth) to their nearby *Shasthya Shebika*.

One field supervisor oversaw the 36 *Shasthya Kormi,* recorded field activities, and conducted follow-up interviews with 5% of participants to confirm visits and measure protocol adherence.

### 2.4. Outcomes

Primary outcomes were the mean birth weight of live-born infants and the proportion of newborns with a birth weight less than 2500 g. The *Shasthya Kormi* had many years of experience measuring birth weight, but we retrained them with their standard equipment, a 5 kg Salter spring balance scale with an accuracy of 100 g. They calibrated the scales with a standard weight every week. We also checked the scales monthly during refresher training. For mothers birthed in health facilities, we recorded the birth weight measured at the facility.

For secondary outcomes, the *Shasthya Kormi* used structured forms to record fetal loss during pregnancy, birth outcome, delivery information, and neonatal or maternal death. They also used monthly semi-structured questionnaires to collect dietary data, with the food list adapted from Food and Nutrition Technical Assistance II Project (FANTA-2) [24]. These questionnaires captured information about foods, beverages, and supplements consumed 24 h before the interview. We did not collect information on the frequency or the quantity of food consumed. From the data collected, we formed seven food groups as follows: (1) grains, white roots and tubers; (2) pulses (beans, peas and lentils), nuts and seeds; (3) dairy; (4) meat, poultry and fish; (5) eggs; (6) Vitamin A-rich fruits and vegetables; (7) other fruit and vegetables. We calculated the mean number of food groups consumed to assess diet quality [25] and compared the mean number of food groups between the treatment groups and the relative differences for each food group. We defined animal source protein as consuming any combination of dairy, meat, poultry, fish or egg food groups.

### 2.5. Statistical Analysis

We required 720 pregnant women to achieve 80% power to detect a difference of 100 g in birth weight between groups. We assumed a mean birth weight of 2531 g (standard deviation (SD) 415 g) in the control group based on the findings from a similar trial in Bangladesh [26]. We set a significance level of 0.05 (two-sided) and assumed an intra-cluster correlation coefficient (ICC) of 0.03 [27] and 36 clusters with an average of 10–15 live births per cluster per month. To adjust for 10% pregnancy loss and 15% loss to follow-up based on similar trials [26,28], we inflated the required sample size to 900 pregnant women.

We analysed the data with Stata version 17 and conducted intention-to-treat analyses. We assigned the intervention randomly at the cluster level but assessed outcomes at the individual level. We estimated the mean between-group difference in birth weight using linear regression and the relative risk for LBW using binomial regression. We included the study group as a covariate and adjusted for cluster randomisation using Generalised Estimating Equations (GEE) [29]. We estimated the intervention effect on maternal dietary diversity and consuming individual food groups in the last 24 h using multilevel linear and Poisson mixed models. The linear mixed model estimated the difference in the mean number of food groups consumed (out of 7), and the Poisson model estimated the relative risk of consuming individual food groups between intervention and control arms. In each model, we included study arms (intervention = 1 and control = 0) and gestational months (2–9) as fixed effects with an interaction term between them. We also included two random effect variables for adjusting the cluster RCT design and the repeated measurements at 2–9 gestational months. After each model, we used the post-estimation *lincom* command to estimate the intervention effect at each gestational month.

We conducted post hoc subgroup analyses for the modifying effects of age, education, family income, and place of delivery on the primary outcome, low birth weight. We added an interaction term to assess whether there was a difference in intervention effect for any sub-group, using a significance level of 0.01.

## 3. Results

We identified 2154 newly pregnant women from August 2015 to February 2016 in the selected clusters. Of these women, 937 did not meet the eligibility criteria (603 had a gestation of >12 weeks, 147 were not permanent residents, and 187 had planned to move out of the study area for childbirth), and 324 declined to participate. We recruited 893 participants, 445 in 18 intervention clusters and 448 in 18 control clusters, and we lost 44 participants during follow-up, 25 (5.6%) intervention, and 19 (4.2%) control. Birth weight was missing for 22 participants (Figure 1). There were 42 pregnancy losses (abortion or stillbirth) and six neonatal deaths within 24 h of birth.

There were no important differences in baseline characteristics between the groups (Table 1). There were small differences in the mother’s age, mother’s and father’s education, and parity but not enough to warrant an adjustment in analyses. Slightly more intervention participants received four or more ANC visits, but this trend reversed for antenatal care from health centres (Table 1). When we combined both ANC data sources, the intervention participants received slightly more ANC (intervention 61.7% vs. control 55.6%). A somewhat higher percentage of intervention participants delivered in a facility (20.7% vs. 16.6%), but this variable reflects the small differences in ANC. Mean birth weight was higher in facility deliveries (2915 g, SD 560 g) than in-home births (2769 g, SD: 410 g).

Supervisors attended 5% of balanced plate demonstrations to observe the appropriateness of message delivery, the use of the menu and food chart, demonstration, problem-solving, and counselling. Approximately 6–7 of 8 messages were delivered to 80% of the participants. The time taken for the demonstrations gradually decreased with increasing numbers of visits. Counselling family members largely depended on availability; husbands were present at 15–20% of visits, and other family members at >80% of visits.

We found a statistically significant difference in mean birth weight and LBW (Table 2 and Table 3). The mean difference in birth weight between groups was 127.5 g (95% CI: 11.1, 243.9; *p* = 0.032) adjusted for clustering. The estimated relative risk (RR) of LBW was 57% lower in the intervention group (RR: 0.43; 95% CI: 0.25, 0.75; *p* = 0.003) adjusted for clustering.

We conducted subgroup analyses of the mother’s age, education, and family income (Table 2 and Table 3) and found a significant intervention effect on birth weight in adolescents vs. non-adolescents. The mean difference in birth weight between groups was greater in adolescents (299.1 g; 95% CI: 101.6, 496.6; *p* = 0.003; vs 95.6 g; 95% CI: −17.3, 208.4; *p* = 0.097) (*p*-value for interaction 0.009) as was the reduction in LBW (RR: 0.28; 95% CI: 0.11, 0.71; *p* = 0.007 vs RR: 0.54; 95% CI: 0.29, 0.98; *p* = 0.044), although this was not statistically significant (*p*-value for interaction 0.22). There was no evidence of treatment modification by the mother’s education or family income (Table 2 and Table 3).

To assess bias in ascertaining birth weight, we stratified by participants birthed in a facility whose babies were weighed by independent observers and those birthed at home and found no evidence of treatment modification (*p* = 0.78).

After adjustment for ANC, using iron and folic acid, and clustering, the reduction in mean birth weight and LBW was smaller in the intervention group (Appendix A Table A1 and Table A2). The adjusted difference was 121.8 g (95% CI: 11.7, 231.9; *p* = 0.030), which is 5.7 g less than the unadjusted analysis (Table 2). The adjusted relative risk of LBW was 0.53 (95% CI: 0.29, 0.98; *p* = 0.042), which was smaller than the unadjusted relative risk in Table 3.

The mean number of food groups consumed increased in the intervention group and was significantly higher from the third month of pregnancy (Figure 2). The largest difference was in the eighth month of pregnancy (Table 4, Figure 2). The mean number of animal-source protein foods consumed was significantly higher from the fourth month of pregnancy. In the fourth gestational month, the intervention participants consumed 1.5 animal-source foods/day (95% CI: 1.38, 1.64) compared to 1.1 (95% CI: 1.02, 1.25) in control participants. The largest difference was in the eighth gestational month when intervention participants consumed 1.9 animal source foods/day (95% CI: 1.68, 2.15) compared to 1.1 (95% CI: 1.02, 1.24) in control participants.

The relative consumption rate of beans, legumes and nuts, dairy food and other fruits and vegetables was significantly higher in the intervention than in the comparison group after the third or fourth gestational month (Table 4). The largest relative increases were for eggs, dairy foods and other fruits and vegetables after the fourth or fifth gestational month (Table 4).

## 4. Discussion

We demonstrated that antenatal nutrition education with practical demonstrations of preparing a balanced diet increases infant birth weight and reduces LBW in rural Bangladesh. After adjustment for ANC, using iron and folic acid, and clustering, there was a higher mean birth weight in the intervention group of public health significance (121.8 g) than in the control group. The effect of the intervention was greater in adolescent mothers. The findings provide strong evidence of the effectiveness of a sustainable intervention suitable for implementation in community-based healthcare systems. They demonstrate the potential impact of nutrition education to improve intrahousehold food distribution as a tool to reduce low birth weight in Bangladesh and across South Asia.

The process evaluation of the trial [30] provided insights into why the intervention was so impactful. It identified key elements in the intervention strategy that was crucial in achieving the desired adherence. These include the practical demonstration of portion sizes, addressing local food perceptions, demystifying animal-source foods, engaging husbands and mothers-in-law, leveraging women’s social networks, and harnessing community health workers’ social roles. In particular, the involvement of the mothers-in-law and husbands changed their attitudes towards a fairer food allocation to pregnant women. The husbands adhered to the messages and purchased more nutritious foods for their wives by adjusting the shopping budget and even working extra hours.

Our intervention appears more effective than interventions using micronutrients or food supplements. A large cluster RCT (22,405 pregnancies) in Bangladesh assessed the impact of antenatal micronutrient supplementation and found a 54 g (95% CI: 41, 66) increase in birth weight and a 12% reduction (RR: 0.88; 95% CI: 0.85, 0.91) in LBW [26]. Another cluster RCT of lipid-based micronutrient supplementation showed a 41 g birth weight increment without a significant effect on LBW [28].

Providing micronutrient supplements, even with additional food energy, is unlikely to address the fundamental problems of insufficient food intake in pregnancy related to food taboos and inequitable intra-household food distribution. The earlier MINIMat RCT (3267 singleton newborns) [31], which tested early and late start to food supplements with micronutrients in pregnancy, found no significant main effects on birth weight. However, combining multiple micronutrients and early food supplementation improved infant survival. The lack of a control group without food supplements might explain the lack of birth weight response. In contrast, an observational study in Bangladesh reported a 118 g increase in birth weight with daily 608 kcal food supplements combined with iron-folic acid sustained for more than four months [32]. Although this evidence is weak, it supports using food-based solutions to address undernutrition in pregnancy.

Adolescent pregnancy increases the risk of adverse birth outcomes, including LBW [33]. If severely undernourished, the growing pregnant adolescent and foetus compete for nutrients [34]. A recent study from rural Bangladesh found a cessation of linear growth, weight loss, and fat and lean body mass depletion in pregnant adolescents [34]. This greater nutritional susceptibility helps explain the greater effect of balanced plate education in this group.

The balanced-plate intervention significantly increased maternal dietary diversity (Table 4, Figure 2). Change in dietary patterns is the likely cause of the higher birth weight in the intervention group. A recent cohort study [35] followed 374 women from their first antenatal care visit to assess associations between dietary diversity during pregnancy and LBW in rural Ethiopia. The LBW risk doubled with a dietary diversity score of less than four food groups (ARR: 2.06; 95% CI: 1.03, 4.11). A facility-based case-control study [36] from rural Ethiopia, which examined iron, folic acid and nutrition counselling, snack consumption, dietary diversity, and maternal undernutrition, also reported similar findings. 

There was a significant six-fold increase in the adjusted odds of LBW with an inadequate minimum dietary diversity score (AOR: 6.65; 95% CI, 2.31, 19.16). A cross-sectional study [37] from Ghana also reported a protective effect of higher dietary diversity scores for LBW (AOR: 0.10; 95% CI: 0.04, 0.13 per standard deviation change score, *p* = <0.0001). This evidence supports our findings that an improved and more diverse dietary intake reduces LBW.

Participants in the intervention group consumed more animal-sourced protein foods, reaching a peak at eight months of gestation of approximately two animal-sourced foods/day. A large cross-sectional study of pregnant women in Shaanxi province, northwestern China, reported an average weekly consumption of five animal-sourced foods [38]. As the frequency of animal-sourced food increased by one time/week, the average birth weight increased by 3.24 g (95% CI: 1.09, 5.39). Unlike increasing animal-sourced foods in the diet, consuming excess protein supplements (>20% of energy as protein) can harm fetal growth [39].

Despite earlier evidence of the positive impact of nutrition education on pregnancy outcomes, the delivery method has received little attention. It is particularly important in low-literate and poor self-efficacy populations, such as rural Bangladesh, where women require dietary counselling to be accessible, feasible, and sanctioned by the community. A review of over 300 studies found that nutrition education is more likely to be effective when focusing on behaviour and action than knowledge alone. And it is also expected to be more effective if grounded in theory [40]. Fishbein and Yzer note that “any given behaviour is most likely to occur; if one has a strong intention to perform the behaviour; if a person has the necessary skills and abilities required to perform the behaviour; and if no environmental constraints are preventing the behavioural performance” [41]. To positively impact dietary behaviour, we designed our intervention to embrace the three key components of this theory. The verbal communication increases awareness and enhances the pregnant woman’s motivation and intention to adopt the promoted dietary behaviours; practical demonstration facilitates the skills to practice balanced meals; and communication with influential family members (husband, mother-in-law and mother) provides an enabling environment to support the woman’s changed dietary behaviour. A similar nutrition counselling intervention increased age-specific complementary feeding practices in rural Bangladesh [42].

In Bangladesh, the main source of nutrition education for pregnant women is through ANC, but the contact time is insufficient to affect nutrition change [43]. Thus, WHO. Proposes that upskilling CHWs in delivering dietary advice is an appropriate approach to complement health services [44]. Our study integrated the balanced plate intervention into BRAC’s existing ANC package delivered by CHWs. The same staff in a recent intervention demonstrated their successful counselling on breastfeeding and complementary feeding [45]. Previously, we also found that CHWs can influence family decision-makers to increase pregnant women’s iron-folic acid consumption [46].

The evidence presented from this trial is robust. Key strengths are the cluster, randomised controlled trial design with sufficient statistical power to detect clinically important changes in outcomes. The delivery of the trial treatments through an established home-based antenatal care service led to a balance between the treatment groups in the level of antenatal contact with community health workers. The unblinded intervention was counterbalanced by objective outcomes and blinding participants and data collectors to the aims and hypotheses. The trial had a very high follow-up (93%), and we used intention-to-treat analyses and adjusted for unbalanced baseline factors and clustering. Our intervention was logistically simple and culturally appropriate based on well-designed qualitative formative research. The key to our success was conducting the trial within existing ANC frameworks. The delivery strategy resonates with maternal and child nutrition recommendations to achieve adequate coverage of nutrition-specific interventions by reaching needy populations [47]. 

However, we acknowledge the study’s limitations. First, *Shasthya Kormi*, who delivered the intervention, collected birth weight data. Measuring birth weight and providing nutrition counselling are routine tasks for these staff. By blinding the SK to the study outcome, we would expect non-differential bias. Also, we found no difference in the effects between women who had birth weight measured by hospital staff versus those who were birthed at home with birth weight measured by the SK. Secondly, using catchment areas as clusters might introduce contamination, with the possibility of *Shasthya Kormi* exchanging information. However, this would have biased results towards a null effect. To prevent this, we trained the intervention and control group *Shasthya Kormi* separately at regular refresher sessions. Thirdly, we could not quantify food intake and intra-household food distribution, which might have provided further evidence of pathways to the observed impact. Fourthly, we could not assess the modifying effects of seasonality on the response to the intervention because we did not collect detailed information about seasonality or food security. Further, the study recruitment period was short, and few women were impacted by seasonal food insecurity. Fifthly, we could not measure pre-pregnancy BMI, which is a potential confounder and might also modify the responses to the nutrition education intervention. Also, we did not measure the mother’s gestational weight gain, which might have provided more information about mechanisms leading to improved birth weight in the intervention arm. Finally, we excluded over 30% of pregnant women who identified because they were already in the second or third trimester when we detected their pregnancy. There is no information about the characteristics of these excluded women, and thus no way to compare them to those included in the trial. This lack of data limits our assessments of the external validity of the trial findings.

## 5. Conclusions

Our results indicate that a balanced plate nutrition education during pregnancy increases birth weight and reduces LBW. Our trial offers a pragmatic food-based strategy for resource-poor environments, especially where there is gender-biased intrahousehold food allocation. The CHW-led intervention is simple, feasible, and easy to implement in settings with an existing community nutrition outreach program. The intervention can be implemented in Bangladesh through the existing community health infrastructure.

## Figures and Tables

**Figure 1 nutrients-14-04687-f001:**
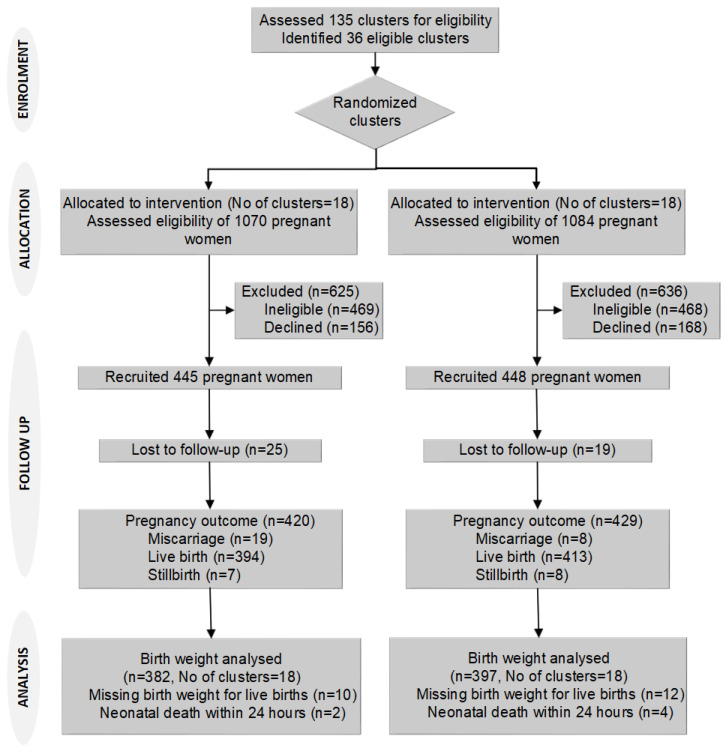
Trial flow chart.

**Figure 2 nutrients-14-04687-f002:**
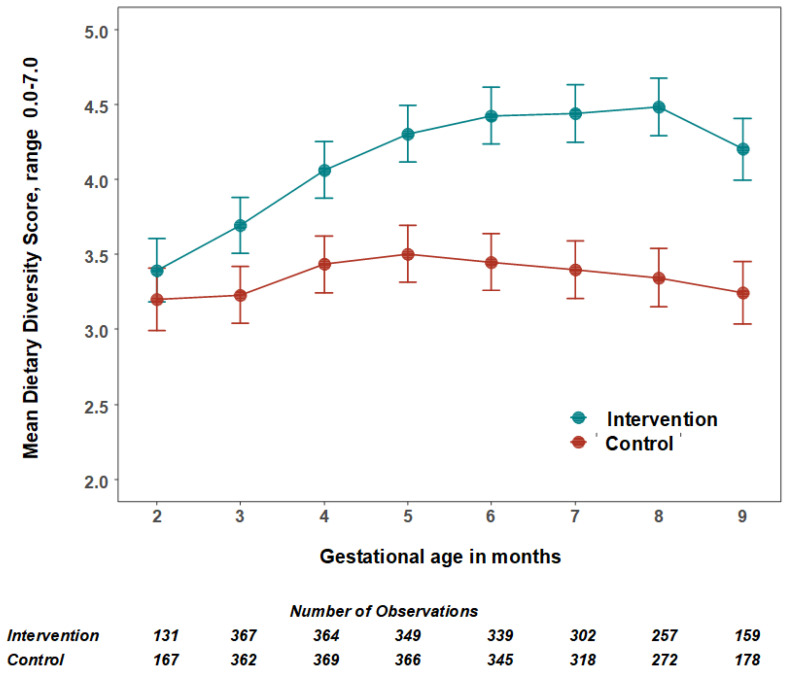
The mean number and 95% confidence intervals of food groups consumed by women per day by treatment group. The estimated mean number of food groups consumed per 24 h, and 95% confidence intervals were calculated using multilevel linear mixed models with study arms (intervention = 1 and control = 0) and gestational months (2–9) as fixed effects with an interaction term between them and two random effect variables for adjusting the cluster RCT design and the repeated measurements.

**Table 1 nutrients-14-04687-t001:** Cluster, baseline maternal and household, and antenatal care and delivery characteristics by study arm.

Characteristics		Intervention (N = 382) *n* (%)	Control (N = 397) *n* (%)
Cluster characteristics			
Number of clusters		18	18
Mean (SD) cluster population in 2016		9775 (2007)	9639 (1763)
Mean (SD number of households per cluster		2315 (288)	2321 (419)
Mean (SD) number of reproductive-age women		2639 (542)	2631 (481)
Mean (SD) pregnant women enrolled per cluster		24.7 (1.0)	24.9 (0.5)
Baseline maternal and household characteristics			
Mother’s age (years)			
Mean (SD)		23.0 (4.6)	24.0 (4.3)
15–19		80 (21.7)	64 (17.0)
20–34		280 (76.1)	307 (81.4)
>34		8 (2.2)	6 (1.6)
Missing		14	20
Mother’s education			
No education		82 (22.0)	89 (23.1)
Primary		250 (67.2)	242 (62.7)
Secondary		27 (7.3)	35 (9.1)
Higher		13 (3.5)	20 (5.2)
Missing		10	11
Monthly family income (SD) in Bangladesh Taka		7308 (4303)	7213 (4438)
	Less than 5000	87 (23.9)	79 (21.2)
	5000 to less than 10,000	190 (52.2)	209 (56.2)
	10,000 or more	87 (23.9)	84 (22.6)
	Missing	18	25
Father’s education			
	No education	130 (35.0)	151 (39.7)
	Primary	188 (50.7)	169 (44.5)
	Secondary	34 (9.2)	39 (10.3)
	Higher	19 (5.1)	21 (5.5)
	Missing	11	17
Marital Status			
	Married	372 (97.4)	386 (97.3)
	Not married	10 (2.6)	11 (2.7)
Parity			
Mean (SD)		1·8 (1.2)	1·9 (1.3)
	0	36 (10.3)	48 (12.6)
	1	134 (38.5)	115 (30.3)
	≥2	178 (51.2)	217 (57.1)
	Missing	34	17
Previous pregnancy loss			
	Yes	67 (19.3)	87 (22.9)
	No	281 (80.8)	293 (77.1)
	Missing	34	17
Gestation at enrolment (weeks)	Mean (SD)	10.1 (3.2)	9.6 (3.2)
	Missing	27	2
Antenatal care and delivery characteristics			
Iron Folic Acid Supplements			
	Mean number consumed (SD)	45.6 (61.8)	29.0 (46.8)
	Any use	117 (30.6)	70 (17.6)
	None	265 (69.4)	327 (83.4)
Antenatal care visit by SK			
	Mean number of visits (SD)	3.8 (2.6)	3.6 (2.2)
	<4 visits	158 (41.1)	198 (49.9)
	≥4 visits	224 (58.6)	199 (50.1)
Antenatal care in health centres			
	Mean number of visits (SD)	0.7 (1.1)	1.0 (1.4)
	No Visit	227 (59.4)	246 (51.9)
	<4 visits	143 (37.4)	169 (42.6)
	≥4 visits	12 (3.1)	22 (5.5)
Place of delivery			
	Facility	79 (20.7)	66 (16.6)
	Home	280 (73.3)	323 (81.4)
	Unknown	23 (6.0)	8 (2.0)
Mode of delivery			
	Normal	331 (86.7)	361 (90.9)
	C-Section	33 (8.6)	34 (8.6)
	Unknown	18 (4.7)	2 (0.5)
Time of birth weight measurement			
	Within 24 h	266 (69.6)	290 (73.1)
	24 to 48 h	75 (19.6)	61 (15.4)
	48 to 72 h	13 (3.4)	19 (4.8)
	Unknown	28 (7.3)	27 (6.8)

Numbers are frequencies (and column percentages) unless stated otherwise.

**Table 2 nutrients-14-04687-t002:** The mean birth weight and the mean difference in birth weight of balanced-plate intervention compared to usual programs.

	Intervention	Control	Intervention	Control	Estimate (95%CI)	*p*-Value	*p*-Value for Interaction
			Mean (SD)	Mean (SD)			
Overall	382	397	2861	2736.8	127.5	0.032	
			(444.2)	(423.2)	(11.1, 243.9)		
Mother’s age ^1^ (years)							0.009
<20	80	64	2916.3	2637.5	299.1	0.003	
			(471.1)	(481)	(101.6, 496.6)		
>=20	288	313	2847.6	2762.6	95.6	0.097	
			(443.4)	(401.1)	(−17.3, 208.4)		
Mother’s education ^2^							0.33
None	82	89	2869.5	2693.3	175.1	0.018	
			(460.6)	(339.7)	(29.9, 320.4)		
Primary	250	242	2839.2	2761.6	92.9	0.132	
			(443.9)	(394.4)	(−28.0, 213.7)		
Secondary or higher	40	55	2982.5	2741.8	226.1	0.088	
			(435.5)	(588)	(−33.5, 485.8)		
Family income ^3^							0.236
Low	87	79	2819.5	2663.3	166.8	0.036	
			(397.9)	(420.4)	(10.7, 322.8)		
Medium	190	209	2806.8	2750.7	51.8	0.404	
			(421.7)	(390.8)	(−70.0, 173.6)		
High	87	84	2995.4	2791.7	183.3	0.109	
			(514.5)	(433.2)	(−40.9, 407.4)		

Birth weight was analysed using linear regression. We adjusted all models for clusters using generalised estimating equation (GEE) models, assuming an exchangeable correlation structure and applied a sandwich estimator to standard errors. ^1^
*n* = 745 due to 34 missing maternal age values. ^2^
*n* = 758 due to 21 missing maternal education values. ^3^
*n* = 736 due to 43 missing family income values.

**Table 3 nutrients-14-04687-t003:** Percentage of low birth weight and relative risk of balanced-plate intervention compared to usual programs.

	Number of Participants		Low Birth Weight (<2500 g)		Relative Risk		
	Intervention	Control	Intervention	Control	Estimate (95%CI)	*p*-Value	*p*-Value for Interaction
			*n* (%)	*n* (%)			
Overall	382	397	37	87	0.43	0.003	
			(10)	(22)	(0.25, 0.75)		
Mother’s age ^1^ (years)							0.217
<20	80	64	6	17	0.28	0.007	
			(8)	(27)	(0.11, 0.71)		
>=20	288	313	31	62	0.54	0.044	
			(11)	(20)	(0.29, 0.98)		
Mother’s education ^2^							0.24
None	82	89	11	22	0.54	0.069	
			(13)	(25)	(0.28, 1.05)		
Primary	250	242	25	47	0.5	0.029	
			(10)	(19)	(0.26, 0.93)		
Secondary or higher	40	55	1	12	0.11	0.029	
			(3)	(22)	(0.02, 0.8)		
Family income ^3^							0.441
Low	87	79	9	25	0.31	0.01	
			(10)	(32)	(0.13, 0.76)		
Medium	190	209	23	41	0.6	0.117	
			(12)	(20)	(0.32, 1.14)		
High	87	84	5	12	0.45	0.199	
			(6)	(14)	(0.13, 1.53)		

Low birth weight was analysed using binomial regression with a log link. We adjusted all models for clusters using generalised estimating equation (GEE) models, assuming an exchangeable correlation structure and applied a sandwich estimator to standard errors. ^1^
*n* = 745 due to 34 missing maternal age values. ^2^
*n* = 758 due to 21 missing maternal education values. ^3^
*n* = 736 due to 43 missing family income values.

**Table 4 nutrients-14-04687-t004:** Effect of balanced plate nutrition education intervention on maternal dietary diversity and individual food groups.

		Maternal Dietary Diversity and Individual Food Groups							
		Dietary Diversity	Starchy Food	Beans, Legumes, Nuts	Dairy Foods	Fish and Meat	Eggs	Vitamin A-Rich Foods	Other Fruits and Vegetables
		Mean Diff. (95% CI)	RR (95% CI)	RR (95% CI)	RR (95% CI)	RR (95% CI)	RR (95% CI)	RR (95% CI)	RR (95% CI)
Months of Pregnancy	2	0.19	1.16	1.19	2.31	1.12	1.48	1.17	1.92
		(−0.11, 0.49)	(0.95, 1.41)	(0.82, 1.75)	(1.09, 4.91)	(0.9, 1.39)	(0.72, 3.04)	(0.7, 1.96)	(0.80, 4.63)
	3	0.47	1.01	1.37	1.44	1	1.97	1.12	2.59
		(0.20, 0.73)	(0.99, 1.03)	(1.06, 1.76)	(0.78, 2.66)	(0.97, 1.04)	(1.00, 3.88)	(0.72, 1.75)	(1.24, 5.43)
	4	0.63	1.03	1.4	1.69	1.07	2.51	1.15	2.38
		(0.36, 0.90)	(0.99, 1.08)	(1.11, 1.77)	(0.82, 3.51)	(0.99, 1.14)	(1.28, 4.94)	(0.87, 1.52)	(1.19, 4.75)
	5	0.80	1.02	1.38	2.36	1.05	2.92	1.2	2.87
		(0.53, 1.07)	(0.98, 1.07)	(1.07, 1.79)	(1.10, 5.08)	(0.97, 1.13)	(1.59, 5.36)	(0.92, 1.57)	(1.34, 6.14)
	6	0.98	1.02	1.5	2.81	1.06	3.59	1.26	2.88
		(0.71, 1.25)	(0.99, 1.06)	(1.17, 1.93)	(1.35, 5.85)	(1.00, 1.13)	(2.00, 6.41)	(0.98, 1.62)	(1.32, 6.25)
	7	1.04	1.03	1.66	2.97	1.09	3.59	1.14	3.64
		(0.77, 1.31)	(0.97, 1.08)	(1.29, 2.14)	(1.40, 6.32)	(1.01, 1.17)	(1.91, 6.77)	(0.87, 1.49)	(1.71, 7.73)
	8	1.14	1.05	1.86	2.92	1.06	3.82	1.26	5.34
		(0.86, 1.42)	(0.97, 1.13)	(1.37, 2.52)	(1.54, 5.54)	(0.97, 1.15)	(2.17, 6.73)	(0.95, 1.67)	(2.54, 11.19)
	9	0.96	1.09	1.81	2.34	1.14	4.65	1.37	4.55
		(0.67, 1.25)	(0.92, 1.30)	(1.11, 2.93)	(1.21, 4.55)	(0.95, 1.36)	(2.18, 9.88)	(0.97, 1.94)	(2.04, 10.16)

Inter. = Intervention, Cont. = Control, Mean Diff. = Mean Difference, CI = Confidence Interval, RR = Relative Risk.

## Data Availability

The data described in this manuscript are available on request from the corresponding author.

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
