# Peer review of "The Impact of Antenatal Balanced Plate Nutrition Education for Pregnant Women on Birth Weight: A Cluster Randomised Controlled Trial in Rural Bangladesh"

_nutrients, 2022, doi:10.3390/nu14214687_

Round 1

Reviewer 1 Report

The authors presented interesting and potential important findings that a community-based balanced plate nutrition education intervention effectively increased mean birth weight and reduced LBW, and improved dietary diversity in rural Bangladeshi women. I believe a revision considering following comments could help the authors improve this manuscript.

1. Introduction

a. Line 47-70. Low birth weight is associated with maternal undernutrition. This article introduces malnutrition and low birth weight during pregnancy, followed by health education and birth weight, a little out of place. It is better to add the relationship between dietary health education during pregnancy and nutritional status during pregnancy, and then introduce the research status of dietary health education during pregnancy affecting birth weight.

2. Materials and Methods

a. As for a cluster randomized controlled trial, though many factors have been considered, the design of control group might not be appropriate, both of which pregnant women in the control group received the same frequency of home and standard nutrition advice except for the balanced plate demonstrations. From this aspect, seem both the intervention and the control group receive standard advice affecting dietary behavior, which may deflect the research.

b. Line 116-121. “We used proportionate stratified sampling to randomly select 36 of 135 Shasthya Kormi clusters (4, 5, 7, 13, and 7 from the five sub-districts).”, the detailed proportion should be specified. Moreover, “36 of 135 Shasthya Kormi clusters” is inconsistent with the description in the trial flow chart.

c. Pre-pregnancy BMI, gestational weight gain, sex and number of births, which are closely related to birth weight, are not included in the overall research design. Therefore, these factors related to the outcome variables should be taken into account.

d. Compliance of the experimental and control groups. Whether all the experimental groups could comply with the intervention, and vice versa, which should be detailed. What’s more, the specific implementation process of the control group also needs to be described in detail, otherwise there is difficult to judge that other factors except the intervention factors are as similar as possible.

e. We did not find any information about adjusted variables in the article and the selection of covariates needs to be described in detail.

3. Results

a. Table 1. significances of differences between control and intervention groups need to be tested, and p-values should be added to Table 1.

b. Line 291-295. The amount of each food in the meat group at each stage of pregnancy should be tabulated.

4. Discussion

a. Line382-387. Fishbein and Yzer stated that "any given behavior is most likely to occur; if one has a strong intention to perform the behavior; if a person has the necessary skills and abilities required to perform the behavior; and if no environmental constraints are preventing the behavioral performance". It is not appropriate. In addition, reasons for the balanced plate demonstrations impacting birth weight need to be further discussed.

Reviewer 2 Report

General: Excellent paper precisely written in clear language.

Specific

1.     Table 1: Mother’s age is reported as Mean (SD), but the subgroups are to be corrected  into No. (%)

2.     Line 259: The calculation of the sample size is valid for the comparison of ‘Intervention’ and ‘Control’ only. A multivariate analysis would be appropriate to examine the impact of different confounders (mother's age, education, and family income) to avoid a multiple testing bias.

3.     One concern is raised because of the data shown in figure 2: the curves of both groups flatten and go down towards the nineth month. Did the authors consider seasonal food availability and accessibility ? – If not, this should be mentioned as a limitation.

Reviewer 3 Report

This study assessed the impact of antenatal balanced plate nutrition education for pregnant women on birth weight in rural Bangladesh. It assessed the effect of the intervention via a parallel, two-arm, cluster-randomized controlled trial, with 36 clusters allocated equally to intervention or standard care group. The participants in the intervention group received education about eating balanced meals to meet daily dietary requirements with diverse food groups from their first trimester until delivery. The study showed the mean birth weight was 127.5 g higher in the intervention group compared to the control group, and the intervention reduced the risk of low birth weight (LBW) by 57%. The mean number of food groups consumed was significantly higher in the intervention group from the third month of pregnancy than in the control group. The study demonstrated a community-based balanced plate nutrition education effectively increased mean birth weight, reduced LBW, and improved dietary diversity in rural Bangladeshi women. Overall, the manuscript was well written. There are just some minor questions for authors to clarify.

1.     Please specify the limitations of the existing study --reference [15] in Introduction.

2.     How was the randomization carried out, e.g. by use of random numbers?

3.     Were all pregnant women eligible to the study in the study areas included in the study?

4.     How was sample size identified?

5.     What kind of information did you collect for dietary intake e.g. food group, frequency, and gram?

6. How was the dietary diversity score calculated?
